# Validity and Reliability of the Arabic Version of the Psychosocial Impact of Dental Aesthetics Questionnaire for Yemeni Adolescents

**DOI:** 10.3390/children8060448

**Published:** 2021-05-25

**Authors:** Amal A. M. Alsanabani, Zamros Y. M. Yusof, Wan Nurazreena Wan Hassan, Khalid Aldhorae, Helmi A. Alyamani

**Affiliations:** 1Department of Community Oral Health and Clinical Prevention, Faculty of Dentistry, Universiti Malaya, Kuala Lumpur 50603, Malaysia; dr.amalali88@yahoo.com; 2Department of Preventive Dentistry, Faculty of Dentistry, Sana’a University, Sana’a 2124, Yemen; 3Department of Paediatric Dentistry and Orthodontics, Faculty of Dentistry, Universiti Malaya, Kuala Lumpur 50603, Malaysia; wannurazreena@um.edu.my; 4Department of Orthodontics, Faculty of Dentistry, Thamar University, Thamar 2153, Yemen; drdurai2008@gmail.com; 5Dental Clinics, Kludi Vayan Medical Center, Ministry of Health, Sana’a 2124, Yemen; dr.helmialyamani@yahoo.com

**Keywords:** oral health related quality of life, adolescent, malocclusion, validation

## Abstract

(1) Objectives: This paper aimed to cross-culturally adapt the Psychosocial Impact of Dental Aesthetics Questionnaire (PIDAQ) into an Arabic language version (PIDAQ(A)) for measuring the oral health related quality of life related to dental aesthetics among 12–17-year-old Yemeni adolescents. (2) Material and methods: The study comprised three parts, which were linguistic validation and qualitative interview, comprehensibility assessment, and psychometric validations. Psychometric properties were examined for validity (exploratory factor analysis (EFA), partial confirmatory factor analysis (PCFA), construct, criterion, and discriminant validity) and reliability (internal consistency and reproducibility). (3) Results: The PIDAQ(A) contained a new item. EFA extracted three factors (item factor loading 0.375 to 0.918) comprising dental self-confidence, aesthetic concern, and psychosocial impact subscales. PCFA showed good fit statistics (comparative fit index (CFI) = 0.928, root-mean-square error of approximation (RMSEA) = 0.071). In addition, invariance across age groups was tested. Cronbach’s α values ranged from 0.90 to 0.93 (intraclass correlations = 0.89–0.96). A criterion validity test showed that the PIDAQ(A) had a significant association with oral impacts on daily performance scores. A construct validity test showed significant associations between PIDAQ(A) subscales and self-perceived dental appearance and self-perceived need for orthodontic braces (*p* < 0.05). Discriminant validity presented significant differences in the mean PIDAQ(A) scores between subjects having severe malocclusion and those with slight malocclusion. No floor or ceiling effects were detected.

## 1. Introduction

Malocclusion is a condition of poor dental arrangement and has been shown to correlate with higher levels of dissatisfaction with dental appearance [1,2]. Recent years have seen increased interest in research on patients’ experiences concerning the impact of their dental aesthetics on their oral health related quality of life (OHRQoL) [3,4]. Indices of OHRQoL that measure the subjective perceptions of patients are considered an important complement to normative clinical indicators [5].

Normative assessment of orthodontic treatment need using clinical indices has attempted to analyze the dental health and aesthetic aspects of malocclusion [6]. One of the common indices used in this respect is the Index of Orthodontic Treatment Need (IOTN). It consists of two components: the Dental Health Component (DHC) and the Aesthetic Component (AC) to record the need for orthodontics treatment [7]. However, some indices ignore the degree to which individuals’ perceptions of their own malocclusions have influenced their quality of life and welfare [8]. The differences between professionals’ and individuals’ perceptions of aesthetic effects and orthodontic treatment need are considerable [9]. The psychosocial consequences that may arise from a particular malocclusion should not be disregarded.

The Psychosocial Impact of Dental Aesthetics Questionnaire (PIDAQ) is a self-reported measure that provides information on perceived oral impacts related to dental aesthetics [10]. These problems may result from the presence of dentofacial aesthetic problems; disturbances in oral functions such as speech, mastication, and swallowing; or they may be related to trauma, caries, or periodontal disease [11].

The PIDAQ, originally developed in German, is intended for measuring orthodontic treatment need in persons between 18 and 30 years old seeking orthodontic treatment [9]. The adolescent version for 11- to 17-year-olds has also shown good psychometric properties [12]. The PIDAQ has been adapted in other languages such as Spanish [13,14], Malay [15], Persian [16], and Swedish [17].

The original PIDAQ consists of 23 items grouped into 4 subscales: Dental Self-Confidence (DSC); Social Impact (SI); Psychological Impact (PI); and Aesthetic Concern (AC). The DSC subscale consists of 6 items from the Self-Confidence Scale [9,18]. The SI subscale with eight items is derived from the Social Aspects Scale of the Orthognathic Quality of Life Questionnaire (OQLQ) [19]. The items of the PI subscale were formulated based on the psychological impact of dental aesthetics. The final AC subscale is obtained from the Aesthetics Scale of the OQLQ and consists of 3 items.

In Yemen, government dental services are limited to the provision of fillings, endodontic treatments, extractions, and minor surgeries. Specialized treatments such as orthodontics are only available in dental schools and private clinics. General dental services provided by dental schools for the public are usually free, while patients pay subsidized fees for orthodontic and other specialty treatment [20]. After patients are referred to the respective specialty, they are registered in the waiting list to queue for treatment. Therefore, the demand for orthodontic treatment in Yemen has increased such that in some dental faculties, the waiting list is more than three years long [20]. However, there is no prioritization of treatment, and patients with severe malocclusion are being deprived of immediate treatment and have to wait together with the less severe cases. A self-reported measure of oral impact related to dental aesthetics such as the PIDAQ would be highly useful to compliment clinical indices in assessing the need and treatment priority among patients with malocclusions.

To date, there has been no cross-culturally adapted PIDAQ for the Yemeni population, whose native language is Arabic. Therefore, the aim of this study was to conduct a cross-cultural adaptation of the PIDAQ into an Arabic language version [PIDAQ(A)] for use by 12–17-year-old Yemeni adolescents.

## 2. Materials and Methods

The procedures for cross-cultural adaptation were based on previous related studies [12,15] that were in accordance with standard protocols [21] and recommendations [22,23]. The cross-cultural adaptation comprised linguistic validation and psychometric validation.

The linguistic validation of the PIDAQ consisted of forward translation of the questionnaire into Arabic, back translation, committee review of a PIDAQ(A) draft, and second assessment by appointed evaluators. The PIDAQ(A) draft was then pilot tested for comprehensibility assessment followed by testing of the response format.

Phase 2 comprised the psychometric validation of the PIDAQ(A) involving factor analysis, reliability which is achieved by internal consistency and reproducibility, and validity analysis that involved criterion, construct, and floor and/or ceiling effects.

### 2.1. Linguistic Validation

#### 2.1.1. Forward Translation

Independently, four Yemeni translators whose mother tongue was Arabic and who were proficient in English (a teacher, two professional translators, and an orthodontist) translated the PIDAQ published in English [10] into the Arabic language.

#### 2.1.2. Synthesis of Translations

The four forward translators and the investigator (AAMA) met together and compared the forward translations (T1, T2, T3, and T4) with regard to the content and wording to ensure conceptual and item equivalence between the Arabic version and the source PIDAQ. Finally, a single translated draft of the PIDAQ(A) achieved consensus.

#### 2.1.3. Back Translation

This step was carried out independently by three Yemeni translators who were proficient in English. They had not seen the original PIDAQ when back translating the PIDAQ(A) draft.

#### 2.1.4. Committee Review

An expert committee (EC) consisting of the investigator (AAMA), a Yemeni health professional, a public health expert (SA), and all the forward and back translators met together and compared the three back translations with the original PIDAQ. The EC made critical decisions to attain experiential and semantic equivalences between the source and target versions [24].

#### 2.1.5. Assessment by Appointed Evaluators

The appointed evaluators consisted of two language experts whose task was to confirm the accuracy of the PIDAQ(A) draft. They compared the consensus back translation with the original PIDAQ with regard to semantic equivalence and verified the PIDAQ(A).

### 2.2. Pilot Test: Qualitative Interview

A face-to-face in-depth interview with 30 conveniently sampled 12–17-year-old Yemeni children was conducted by the investigator (AAMA) to test the conceptual and item equivalence of the PIDAQ(A) draft in terms of how oral conditions and dental aesthetics affect the daily lives of Yemeni adolescents, and whether the PIDAQ(A) items were relevant to them. Subjects were also asked general and specific questions about how their dental appearance is affecting their life. The sample of 30 subjects was considered sufficient for this purpose until the accumulated data were exhaustive and no further additional information was obtained. After collecting the items from the qualitative interview and omitting repetitions, these items were compared with the items of the PIDAQ. At this stage, a new item was identified and was consequently added to the PIDAQ(A). The total number of items in the PIDAQ(A) was 24.

### 2.3. Comprehensibility Assessment

This assessment was conducted with 30 12–17-year-old Yemeni school children who were not involved in the pilot test. The aim was to assess the comprehensibility and clarity of the PIDAQ(A), including the format and instructions. The children were asked to answer the PIDAQ(A) and a comprehensibility questionnaire independently. In the comprehensibility questionnaire, the subjects were asked about their understanding of the 24 questions in the PIDAQ(A). The level of comprehensibility of each question was assessed on a four-point rating scale ranging from 0 (not comprehensible) to 3 (highly comprehensible) [25]. Subsequently, the investigator (AAMA) held a discussion with the children on their understanding of the PIDAQ(A), covering the general layout of the questionnaire from the instructions to the questionnaire questions and answer choices. Words that were considered ambiguous were discussed, and suggestions for improvement were sought. The feedback from the subjects was used to further improve the PIDAQ(A).

### 2.4. Testing of Response Format

The PIDAQ(A) was distributed to a convenient sample of 100 12–17-year-old Yemeni adolescents to assess the impact of the response format on the respondents’ performance on the measure. The findings could highlight items with potential problems related to the response format which could influence the overall performance of the index.

### 2.5. Psychometric Validation

A cross-sectional study was carried out with a nonrandom sample of 385 12–17-year-old Yemeni adolescents who were not involved in any of the previous tests. The aim was to assess the psychometric properties of the PIDAQ(A). Subjects were divided into two age groups, 12–14 and 15–17 years. A multistage sampling technique was used to select the sample. First, one district in the city of Sanaa was randomly selected, i.e., the Al-Tahrir district. Next, schools with primary and secondary educations in the Al-Tahrir district were randomly chosen one at a time until the sample size was achieved. Randomization was achieved using simple random sampling.

Altogether, five schools were selected. All 12–17-year-old students from grade 7 to grade 12 were included. Exclusion criteria were subjects who are having or had had orthodontic treatment and subjects with craniofacial deformities.

Subjects completed a questionnaire comprising the PIDAQ(A) and items from the Perception of Occlusion Scale (POS) [26]. They also self-rated their dental appearance using the Aesthetic Component of the Index of Orthodontic Treatment Need (IOTN-AC) [7]. After answering the questionnaire, the subjects were clinically examined to assess malocclusion using the Dental Health Component of the IOTN (IOTN-DHC). IOTN-AC and POS were assessed as well. The subjects were examined by the investigator (AAMA) who had been trained on the use of the IOTN by two orthodontists (WNWH and KA). The inter-examiner reliability was assessed by examining twenty patients, and the process was repeated after two weeks for intra-examiner reliability. Inter- and intra-examiner reliability for both indices (IOTN and POS) was assessed using Kappa scores conducted on 20 subjects. The Kappa scores for inter- and intra-examiner calibration of the IOTN and POS ranged from 0.82 to 0.95. In general, Kappa scores of 0.80 and above 0.90 indicate strong and almost perfect agreement, respectively [27]. In this study, 30% of the subjects were asked to answer the PIDAQ(A) again after 2–3 weeks.

### 2.6. Statistical Analysis

IBM SPSS AMOS v.24 and SPSS v.23 were used for data analysis. Subjects’ agreement to each item of the PIDAQ(A) was measured on a 5-point Likert scale ranging from 1 (never) to 5 (very strongly) [12]. The subscale score was computed by the sum of the scores of its items except for the DSC subscale, where the item scores were reversed before summing up. The overall PIDAQ(A) score was calculated by summing up the subscale scores. Higher PIDAQ scores indicate poorer psychosocial impacts related to dental aesthetics [12].

#### 2.6.1. Factor Analysis

Exploratory factor analysis (EFA) of the 24-item PIDAQ(A) was conducted using principal component analysis (PCA) to extract potential factors [28,29]. In this study, the component correlation matrix of the EFA with Varimax rotation showed values > 0.30, indicating that the factors were correlated. Therefore, Promax rotation was used [30,31]. The Kaiser–Meyer–Olkin measure of sampling adequacy was tested [22].

Estimates of maximum likelihood discrepancy were calculated to determine the partial confirmatory factor analysis (PCFA). Skewness and kurtosis values were assessed to determine data normality [32]. The comparative fit index (CFI) and root-mean-square error of estimation (RMSEA) were used to assess the goodness-of-fit of the observed data to the model. CFI values above 0.90 and close to 1 indicate good fit [33], and RMSEA values between 0.05 and 0.08 show acceptable fit [34]. In addition, measurement invariance between the two age groups was tested using multiple-group PCFA, which examined the change in the goodness-of-fit index (GFI) when cross-group constraints were imposed on a measurement model.

#### 2.6.2. Reliability and Validity

For reliability assessment, the internal consistency of each subscale was tested by measuring the Cronbach’s α, determining where an item was deleted, inter-item correlation, and item–total correlation separately within each of the two age groups for the subscales. For reproducibility, the intraclass correlation coefficient (ICC) was measured. In addition, any significant change between the test and retest administrations was determined using paired *t*-test and Bland and Altman analysis. The limits of agreement were tabulated as mean change ± 1.96× standard deviation of the changes [35]. Concerning validity assessment, concurrent criterion validity was evaluated by measuring the correlation between PIDAQ(A) scores and scores of the Arabic language version of the Child Oral Impacts on Daily Performances (Child-OIDP) index [36]. The Child-OIDP measures oral impact on 8 daily activities, i.e., eating, speaking, and pronouncing properly, cleaning teeth, relaxing and sleeping, emotional status, smiling, doing schoolwork, and socializing. In the Child-OIDP, the impact of malocclusion was accounted for by oral impacts arising from “position of the teeth” and/or “spaces between teeth” [37]. The Child-OIDP performance score was tabulated following past studies [15,38]. The Pearson correlation coefficient was conducted to test the association between PIDAQ(A) scores and Child-OIDP total scores [15].

Convergent validity and discriminant validity were assessed as part of construct validity [39]. Convergent validity was assessed by comparing the questionnaire with other instruments testing related fields, i.e., perceived dental appearance rank and perceived satisfaction with dental appearance rank. Subjects were required to rank their perceived dental appearance by selecting one of the following options: excellent, good, average, or poor. Mann–Whitney statistics was conducted to assess the relationship between the PIDAQ(A) and perceived dental appearance rank. Concerning perceived satisfaction with dental appearance, subjects rated this as very satisfied, satisfied, dissatisfied, or very dissatisfied. The relationship between PIDAQ(A) scores and satisfaction with dental appearance rank was analyzed via Kruskal–Wallis test.

Discriminant validity was assessed by (1) comparing the relationship between PIDAQ(A) scores with self-rated perceived need (MI-S) and investigator-rated need for orthodontic treatment (MI-D), and (2) by comparing its relationship with the IOTN-DHC and with perceived need for braces, i.e., whether or not their teeth needed braces, with response options of “Yes” or “No”. The POS component contained six items of malocclusion traits [26]. Rating the POS component required subjects to assess each item on a five-point Likert scale ranging from “not at all” to “very strongly”. Analysis of the severity of malocclusion using the MI-S and MI-D was adapted from previous studies [12,15]. To compare the relationship between the PIDAQ(A) subscales and the MI-S and MI-D, an independent t-test was applied within each age-group. Effect size estimate was calculated to estimate the clinical significance [40]. The floor and/or ceiling effects were deemed present when more than 15% of subjects achieved the lowest or highest possible score [22].

Ethical approval to conduct this study was obtained from the Medical Ethics Committee, Faculty of Dentistry, Universiti Malaya, Malaysia and from the Faculty of Dentistry, Thamar University, Yemen, prior to taking any steps in the study (see also “Institutional Review Board Statement” below for further information. Permissions were sought from school principals to enlist students for psychometric validation tests prior to commencement of the study.

## 3. Results

The face and content validity of the PIDAQ(A) was confirmed by the experts who concluded that it achieved item and conceptual equivalences with the source version. Any words which upon translation reflected various concepts in the Arabic language and Yemeni culture were replaced with suitable words having similar concepts to the source items. Following the qualitative interview with the 30 subjects, one new item was identified and added to the 23-item PIDAQ(A), which is, “I find myself not attractive because of my teeth”. In the comprehensibility assessment with another 30 subjects, 18 of the 24 items were fully comprehensible (100%). The comprehensibility ratings of the remaining 6 items (items 5, 13, 14, 22, 23, and 24) ranged from 86% (item 5) to 96% (items 14 and 23). Overall, all 30 subjects understood the PIDAQ(A) well with regard to its intent, content, wording, general design, instructions, and response options with minor modifications. Finally, testing of the response format with 100 subjects showed that all were able to answer the PIDAQ(A) satisfactorily.

### 3.1. Psychometric Validation

#### 3.1.1. Descriptive Statistics

Overall, the sample comprised 174 subjects from the first age group (12–14 years old) and 211 from the second age group (15–17 years old). The mean age was 14.7 years (SD = 1.67). The proportions of male and female subjects were 44.2% and 55.8%, respectively. The PIDAQ(A) mean score of the whole sample was 58.0 (SD = 21.7; range = 24–119). The mean score for the first age group was 58.9 (SD = 22.4; range = 24–119), and the mean score for the second age group was 57.3 (SD = 21.2; range = 24–117). No significant difference in PIDAQ(A) mean scores was detected between the older and younger adolescent groups.

#### 3.1.2. Factor Analysis

The Kaiser–Meyer–Olkin measure of sampling adequacy value was 0.96, and the Bartlett’s test was significant (*p* < 0.001). The variables were normally distributed based on the skewness and kurtosis tests [32]. EFA extracted three factors with item loading ranging from 0.375 to 0.918. The first factor comprised 10 items reflecting psychological and social impacts and was named psychosocial impact (PSI). The second factor compromised six items and was similar to the dental self-confidence (DSC) subscale in the original PIDAQ. The third factor comprised eight items reflecting aesthetic concern and was named the aesthetic concern (AC) subscale. The distribution of items in the EFA is shown in Table 1. The three subscales of the PIDAQ(A) explained 65.07% of the total variance.

In the PCFA analysis, the goodness-of-fit analysis revealed good data fit for Model 1 and Model 2 that constrained for the age groups. The results showed that the scores of CFI were in excess of 0.90, and the score of RMSEA was lower than 0.08 with a small confidence interval (Table 2). The factor loadings of the questions under each conceptual variable were within the acceptable range of being over 0.50 for both models A and B. The multigroup PCFA test of the constrained models with the baseline configural model showed invariance across the two age groups (ΔCFI < 0.01) [41,42].

#### 3.1.3. Internal Consistency

The internal consistency, scale statistics, and inter-item correlations of all subscales are presented in Table 3. All subscales achieved Cronbach’s α values above 0.70 for the two age groups. None of the inter-item correlation scores for all subscales were ≥0.90 or ≤0.30. Similarly, the item–total correlation scores were higher than 0.30. Reproducibility test results of the PIDAQ(A) are provided in Table 4. In all subscales, the ICC were greater than 0.80 (*p* < 0.05) in the two age groups. There were no statistically significant differences between test and retest (T1 and T2) excluding the PSI subscale in the 12–14 years old age group. The scores of the smallest detectable change (SDC) were higher in the 12–14 years old age group for all subscales. Bland and Altman plot analysis found that more than 90% of the scores of the second reproducibility test (T2) were within the limits of agreement for all subscales in all age groups.

#### 3.1.4. Validity

##### Concurrent Criterion Validity

Table 5 shows the results of concurrent criterion validity. For all age groups, the PIDAQ(A) scores and Child-OIDP performance scores were significantly associated in all subscales (*p* < 0.01). The three PIDAQ(A) subscales attained statistically higher subscale mean scores for subjects with Child-OIDP impacts. Pearson correlation coefficients revealed that the Child-OIDP performance scores had moderately positive, statistically significant correlations with the DSC, PSI, and AC subscale mean scores (Table 5). The highest correlation coefficient scores were with the DSC subscale: 0.63, 0.61, and 0.61 for the three age groups (12–14, 15–17, and 12–17 years), respectively.

##### Construct Validity

The convergent validity test shows that the associations between the questionnaire subscales and self-perceived dental appearance rank (*p* < 0.01) were statistically significant (Table 6). In all subscales for all age groups, results showed a gradual decrease in the mean scores as subjects appraised their teeth appearance from poor to excellent, and the trend was statistically significant. Ranking of perceived satisfaction with dental appearance showed statistically significant associations with the PIDAQ(A) subscales (*p* < 0.01) (Table 7). For all subscales, the mean scores gradually decreased as subjects ranked their teeth appearance from poor to excellent.

##### Discriminant Validity

Table 8 presents the findings of the discriminant validity test. There was a gradual increase in the mean scores of the PIDAQ(A) subscales with increasing severity of malocclusion, which was represented by the interviewer-rated (MI-D) and self-rated (MI-S) malocclusions. In all subscales (DSC, PSI, and AC), subjects who ranked themselves (MI-S) with no or slight malocclusion and subjects with severe malocclusions (*p* < 0.01) were statistically significantly different. Similarly, there were statistically significant differences between subjects who were ranked by the investigator (MI-D) with no or slight malocclusion and those ranked with severe malocclusions for the three PIDAQ(A) subscales (*p* < 0.01). In all three age groups, comparison between MI-S and MI-D showed that DSC, PSI, and AC subscale mean scores decreased with decreasing severity of malocclusion.

Table 9 shows the associations between the PIDAQ(A) subscales and self-perceived need for braces. Self-perceived need for braces was significantly associated with all subscales of the PIDAQ(A) for all age groups. The IOTN-DHC was significantly associated with the PIDAQ(A) subscales for all age groups (Table 10). The mean PIDAQ(A) scores between subjects who were rated by the investigator with a lower grade of malocclusion and those with a greater grade of malocclusion for all subscales (*p* < 0.01) were statistically significantly different.

##### Floor and/or Ceiling Effects

No floor or ceiling effects were found in any of the age groups (Table 11). As a final corollary of the results, a high PIDAQ can be attributed to the presence of dental aesthetic problems among Yemeni adolescents. 

## 4. Discussion

The guidelines for cross-cultural adaptation of OHRQoL measures were first proposed by Guillemin in 1993 [24]. In this study, the process to cross-culturally adapt the PIDAQ into an Arabic language version followed these guidelines and those by Herdman et al. [23] and Terwee et al. [22]. The cross-cultural steps in this study included assessment of conceptual, item, semantic, and operational measurements, as well as functional equivalences. The final PIDAQ(A) consisted of 24 items, with a new item included which was “I find myself not attractive because of my teeth”.

In this study, the EFA yielded three factors. The first factor was called the psychosocial impact (PSI), which comprised five items from the psychological impact subscale and five items from the social impact subscale of the original PIDAQ. Likewise, the psychosocial impact subscale in the Italian version [43] included the same items from the SI and PI subscales of the original English version except for one item. The second subscale comprised the same items from the dental self-confidence (DSC) subscale of the original English PIDAQ. The DSC subscale in the Arabic, Chinese, and Italian versions had the same items as the original PIDAQ [43,44]. The third factor is called aesthetic concern (AC), which included the same items belonging to the AC subscale of the original PIDAQ in addition to the remaining three items of the social impact and two items of the psychological impact. Similarly, the AC subscale in the Italian version [43] included the same items as the AC subscale in the original version in addition to a fourth item, i.e., “Hold back teeth when smiling”, which is in the SI subscale of the original version. This item is only present in the AC subscale of the Arabic version.

The original version of the PIDAQ is formed out of four subscales [10], just like the majority of the subsequent PIDAQ versions [12,14,15,44,45,46]. However, some other translations have demonstrated a different dimension construction than the original version. Similar to our PIDAQ(A), two other related studies have found that the questionnaire comprised three factors [43,44], while one study yielded only two factors [17], and another study yielded five factors [47]. In addition, there are a small number of adaptations which did not incorporate factor analysis into their validation [48,49,50].

The three factors from the PIDAQ(A) explained 65.07% of the variance, which is slightly higher than the four factors of the original PIDAQ, which explained 63.28% of the variance [10]. In comparing between the findings of factor analysis and the extracted subscales of the Arabic version to the German version, a slight difference was observed. This difference can, to some extent, be attributed to the different cultural backgrounds between the two populations. However, extraction of three subscales from the PIDAQ(A) can represent and reflect information about the impact of psychosocial and dental appearance effects on Yemeni adolescents. Subsequently, the PIDAQ(A) version displayed good psychometric backgrounds and could be used in the local cultural milieu. The PIDAQ(A) has good construct validity, and the content validity was wholly satisfactory. The PCFA showed good data-fit findings and was invariant throughout the age groups, just like the English version for adolescents [12] and the Malay version [15]. The factor loadings showed no difference among age groups. This may lead to the conclusion that the items have similar meanings to all subjects regardless of age, which enables comparing adolescents of different ages.

The Cronbach’s α values were within the acceptable range for all subscales [22,23] and were slightly higher than those of the original version except for the value of the psychological impact, where it was lower. This could be explained by the low number of the PI (three items) in the Arabic version. Likewise, the aesthetic concern subscale in the original version, which included three items, showed the lowest Cronbach’s α value [12]. The ICC scores which assess reproducibility were generally good for all subscales, with scores ranging from 0.89 to 0.96 for the three subscales for the two age groups. These scores were slightly closer to those in the original study, which were between 0.82 and 0.96 [12]. This was similar to the Chinese version where all subscale ICC values of the total subjects were higher than 0.90 [44]. A statistically significant difference was ascertained only in the PSI subscale of the younger age groups, which is similar to the Malay version [15], where a statistically significant difference was observed only in the PI subscale of the younger age groups. Criterion validity was tested by comparing the PIDAQ(A) scores with Child-OIDP score. Criterion validity was evaluated in opposition to the impact of malocclusion on daily activities as evaluated using the Child-OIDP. It also showed statistically significant associations independent of age. The construct validity of the PIDAQ(A) was also assessed by comparing its scores with scores of perceived dental appearance rank and perceived satisfaction with dental appearance rank. All subscales showed statistically significant associations.

Discriminant validity analysis using the self-rated malocclusion (MI-S) and investigator-rated malocclusion (MI-D) revealed statistically significant differences between subjects with no or minor malocclusion and those with significant malocclusion for the three subscales and all age groups. The PSI, DSC, and AC subscale mean scores were higher in those with more severe malocclusion than those with slight malocclusion. The effect sizes were above 0.80 based on the MI-S but were between medium (0.50) and strong (≥0.80) based on the MI-D. These findings were similar to those of previous studies [12,15].

The present study showed no floor and/or ceiling effects for all subscales [22]. Klages et al. [12] showed that increasing values of floor effects were observed with increasing age in the SI, PI, and AC subscales. Similarly, this study recorded a slight increase in floor effects in the three subscales in addition to the DSC subscale.

Finally, the PIDAQ(A) can be used to assess oral impacts related to dental aesthetics among Yemeni adolescents. Future studies should assess its suitability for use in other Arabic populations as well as its evaluative properties to assess changes in OHRQoL following orthodontic treatment. One potential limitation of the present study was the lack of evaluation regarding the responsiveness of the PIDAQ in detecting changes over time as this study has not shown improvement in OHRQoL after applying orthodontic treatment. Nevertheless, this recently developed Arabic version of the PIDAQ can be used in forthcoming studies to evaluate changes in oral-health-related quality of life before and after orthodontic treatment. This could improve our realization and understanding of the responsiveness of the measure, in addition to highlighting the consequences of malocclusions and orthodontic treatment on OHRQoL.

## 5. Conclusions

The current study demonstrated that the English PIDAQ has been successfully cross-culturally adapted into the Arabic language for the Yemeni population with some modifications. This study indicates that the PIDAQ(A) is a valid and reliable instrument to measure oral impacts related to dental aesthetics among adolescents in Yemen. This scale provides information on aspects of OHRQoL related to malocclusion.

## Figures and Tables

**Table 1 children-08-00448-t001:** Factor loadings of the items of the PIDAQ measure (N = 385).

Dimension	Principal Component Analysis		
	Factor 1	Factor 2	Factor 3
Psychosocial Impact (10 items)	Items 24,20,14,5,13,22,6,9,3,11 (Factor loadings = 0.38–0.92)		
Dental Self-Confidence (6 items)		Items 21,4,23,17,12,7 (Factor loadings = 0.71–0.85)	
Aesthetic Concern (8 items)			Items 16,10,1,19,2,15,18,8 (Factor loadings = 0.40–0.95)

Extraction method: principal component analysis. Rotation method: Promax with Kaiser normalization.

**Table 2 children-08-00448-t002:** The multigroup PCFA test of the constrained models with the baseline configural model.

	MODEL 1	MODEL 2	
		(Baseline Configural Model)	
N	312	137	175
**Age-group**	12–17 years	12–14 years	15–17 years
**Fit Indices**			
CFI	0.928	0.906	
RMSEA	0.071 (0.064–0.78)	0.058 (0.053–0.063)	
**Items in Brief**	**Factor loading**	**Factor loading**	
DSC			
4. Proud of own teeth	0.80	0.85	0.77
7. Like to show their teeth	0.69	0.70	0.68
12. Pleased to see own teeth in mirror	0.79	0.79	0.80
17. Teeth look nice to others	0.69	0.71	0.68
21. Satisfied with own teeth’s appearance	0.85	0.86	0.85
23. Find own teeth nice	0.85	0.86	0.85
PSI			
11. Others have nicer teeth	0.64	0.66	0.63
3. Envy others for their teeth	0.70	0.68	0.74
9. Teasing	0.70	0.80	0.61
6. Distressed because of others’ nice teeth	0.76	0.78	0.74
22. Boys/girls find own teeth ugly	0.83	0.84	0.81
13. People look strange at my teeth	0.78	0.85	0.71
5. What others think	0.65	0.67	0.61
14. Shy because of own teeth	0.86	0.89	0.84
20. Wish to look better	0.67	0.58	0.74
24. Not attractive because of own teeth	0.87	0.84	0.88
AC			
8. Don’t like own teeth on photos	0.67	0.67	0.67
18. Don’t like own teeth on videos	0.77	0.79	0.76
15. Hiding own teeth	0.72	0.75	0.79
2. Hold back their smile	0.71	0.69	0.74
19. Stupid comments from others	0.56	0.59	0.54
1. Don’t like own teeth in mirror	0.80	0.80	0.80
10. Unhappy about own teeth	0.79	0.83	0.76
16. Feel bad about own teeth	0.89	0.89	0.90

CFI: comparative fit index. RMSEA: root-mean-square error of approximation (90% CI).

**Table 3 children-08-00448-t003:** Reliability statistics of the PIDAQ(A) subscales.

PIDAQ Subscale	N	Cronbach’s α	Scale Statistics			Inter-Item Correlations		Item–Total Correlation		Cronbach’s α if Item Deleted
			Mean	SD	Mean	Min	Max	Min	Max	
12–14 years										
**DSC**	174	0.91(0.87–0.92)	19.21	6.30	3.20	0.48	0.77	0.65	0.81	0.88–0.90
**PSI**	174	0.93(0.86–0.93)	22.71	10.01	2.27	0.37	0.79	0.56	0.84	0.91–0.92
**AC**	174	0.91(0.88–0.93)	17.01	7.82	2.13	0.38	0.73	0.53	0.80	0.89–0.91
15–17 years										
**DSC**	211	0.90(0.88–0.92)	19.23	6.41	3.21	0.52	0.80	0.66	0.80	0.88–0.90
**PSI**	211	0.92(0.85–0.92)	21.94	9.37	2.19	0.35	0.73	0.59	0.80	0.90–0.92
**AC**	213	0.91(0.88–0.92)	16.22	7.12	2.03	0.34	0.72	0.52	0.84	0.88–0.91
12–17 years										
**DSC**	385	0.90(0.88–0.91)	19.22	6.35	3.20	0.52	0.78	0.66	0.80	0.88–0.90
**PSI**	385	0.92(0.86–0.92)	22.29	9.66	2.23	0.38	0.74	0.62	0.81	0.91–0.92
**AC**	385	0.91(0.89–0.92)	16.58	7.45	2.07	0.36	0.73	0.52	0.81	0.89–0.91

SD: standard deviation.

**Table 4 children-08-00448-t004:** The reproducibility tests of the PIDAQ(A) (test–retest reliability).

PIDAQ	ICC Agreement			Paired T-Test		Bland and Altman		
Subscale	(95% CI)	(SEM)	SDC	MDiff	(SD)	95% Limits of Agreement		
						Lower	Upper	%Within Limits
12–14 years (40)								
**DSC**	0.89 (0.78–0.94)	2.53	7.01	−1.03	(3.59)	−8.06	6.01	95.0
**PSI**	0.91 (0.81–0.95)	3.60	9.98	−2.05 *	(5.10)	−12.05	7.95	97.5
**AC**	0.92 (0.85–0.96)	2.99	8.29	−0.63	(4.24)	−8.47	6.97	90.0
15–17 years (76)								
**DSC**	0.91 (0.85–0.94)	2.23	6.18	−0.30	(3.16)	−6.50	5.90	96.1
**PSI**	0.93 (0.89–0.96)	3.02	8.37	−0.21	(4.27)	−8.58	8.16	96.1
**AC**	0.96 (0.94–0.97)	1.63	4.51	0.07	(2.31)	−4.47	4.60	93.4
12–17 years (116)								
**DSC**	0.90 (0.85–0.93)	2.34	6.49	−0.55	(3.32)	−7.06	5.95	96.6
**PSI**	0.92 (0.88–0.94)	3.28	9.09	−0.84	(4.64)	−9.94	8.25	94.8
**AC**	0.94 (0.92–0.96)	2.20	6.10	−0.17	(3.12)	−6.29	5.94	93.1

* *p* < 0.05 (paired-sample T test). CI: confidence interval. SEM: standard error of measurement. SDC: smallest detectable change. MDiff: mean differences.

**Table 5 children-08-00448-t005:** Concurrent criterion validity of the PIDAQ(A).

PIDAQ Subscale	Child-OIDP Prevalence	N	Mann–Whitney U						Pearson Correlation	
			PIDAQ Scores					*p* Value	Child-OIDP Performance	*p* Value
			Mean	SD	Quartiles					
					Lower	Middle	Upper			
12–17 years										
**DSC**	No	200	14.98	5.09	11.00	14.00	19.00	0.000 **	0.61	0.000 **
	Yes	185	23.81	3.93	22.00	24.00	27.00			
**PSI**	No	200	15.97	4.99	12.00	15.00	18.00	0.000 **	0.58	0.000 **
	Yes	185	29.12	8.80	22.00	29.00	35.00			
**AC**	No	200	11.99	3.01	10.00	12.00	14.00	0.000 **	0.55	0.000 **
	Yes	185	21.54	7.64	15.00	19.00	28.00			
15–17 years										
**DSC**	No	112	15.09	5.43	11.00	14.00	20.00	0.000 **	0.61	0.000 **
	Yes	99	23.91	3.59	22.00	24.00	27.00			
**PSI**	No	112	15.89	5.06	12.00	14.50	19.75	0.000 **	0.58	0.000 **
	Yes	99	28.79	8.37	23.00	29.00	35.00			
**AC**	No	112	11.97	3.17	10.00	12.00	14.00	0.000 **	0.52	0.000 **
	Yes	99	21.03	7.31	15.00	19.00	27.00			
12–14 years										
**DSC**	No	88	14.84	4.66	11.00	14.00	18.00	0.000 **	0.63	0.000 **
	Yes	86	23.69	4.32	21.00	24.00	27.00			
**PSI**	No	88	16.06	4.93	13.00	15.00	18.00	0.000 **	0.60	0.000 **
	Yes	86	29.52	9.29	22.00	29.00	36.00			
**AC**	No	88	12.01	2.81	10.00	11.50	14.00	0.000 **	0.59	0.000 **
	Yes	86	22.12	8.01	15.00	19.50	30.00			

Scores of the DSC subscale items were reversed. ** *p* value < 0.001 for all subscales with all age groups.

**Table 6 children-08-00448-t006:** Convergent validity: association between the PIDAQ(A) and appearance rating.

PIDAQ Variables	Rate Appearance	N	PIDAQ Scores					*p* Value
			Mean	SD	Quartiles			
					Lower	Middle	Upper	
12–14 years								
**DSC**	Excellent	48	13.02	4.03	10.00	13.00	15.75	0.000 **
	Good	57	18.18	4.60	13.50	19.00	22.00	
	Average	43	22.51	3.99	20.00	23.00	25.00	
	Poor	26	27.46	2.16	26.00	27.00	29.25	
**PSI**	Excellent	48	14.48	3.68	12.00	13.00	16.00	0.000 **
	Good	57	19.42	5.62	15.00	18.00	22.50	
	Average	43	26.02	7.16	21.00	26.00	31.00	
	Poor	26	39.65	6.25	34.00	39.00	45.25	
**AC**	Excellent	48	11.63	2.47	10.00	11.00	13.75	0.000 **
	Good	57	14.28	4.65	10.00	14.00	16.00	
	Average	43	18.40	6.01	14.00	17.00	20.00	
	Poor	26	30.62	5.86	28.00	31.00	34.00	
15–17 years								
**DSC**	Excellent	52	11.63	4.24	9.00	11.00	14.00	0.000 **
	Good	58	18.02	4.54	14.00	18.00	21.00	
	Average	66	22.26	3.17	20.75	22.00	24.00	
	Poor	35	26.80	2.31	26.00	27.00	28.00	
**PSI**	Excellent	52	13.44	3.04	11.00	12.50	15.00	0.000 **
	Good	58	17.52	5.41	13.00	16.50	21.25	
	Average	66	25.18	6.82	20.75	25.00	29.25	
	Poor	35	35.80	5.35	32.00	35.00	40.00	
**AC**	Excellent	52	10.46	2.15	9.00	10.00	12.00	0.000 **
	Good	58	13.19	3.23	11.00	13.00	15.00	
	Average	66	17.44	5.41	13.00	17.00	21.25	
	Poor	35	27.51	5.73	25.00	28.00	30.00	
12–17 years								
**DSC**	Excellent	100	12.30	4.18	9.00	12.00	14.00	0.000 **
	Good	115	18.10	4.55	14.00	19.00	22.00	
	Average	109	22.36	3.50	20.00	23.00	25.00	
	Poor	61	27.08	2.25	26.00	27.00	29.00	
**PSI**	Excellent	100	13.94	3.39	11.00	13.00	16.00	0.000 **
	Good	115	18.46	5.57	14.00	18.00	22.00	
	Average	109	25.51	6.94	21.00	25.00	30.00	
	Poor	61	37.44	6.02	32.00	36.00	43.00	
**AC**	Excellent	100	11.02	2.37	9.00	10.00	13.00	0.000 **
	Good	115	13.73	4.01	11.00	13.00	16.00	
	Average	109	17.82	5.65	14.00	17.00	21.00	
	Poor	61	28.84	5.94	25.00	29.00	33.00	

Scores of the DSC subscale items were reversed. ** *p* value < 0.001 for all subscales with all age groups.

**Table 7 children-08-00448-t007:** Convergent validity: association between the PIDAQ(A) and perceived satisfaction with dental appearance.

Subscale	Rate Satisfaction	N	PIDAQ Scores					*p* Value
			Mean	SD	Quartiles			
12–14 years					**Lower**	**Middle**	**Upper**	
**DSC**	Very satisfied	36	12.72	3.95	10.00	12.00	15.75	0.000 **
	Satisfied	62	17.00	4.63	13.00	17.00	21.00	
	Dissatisfied	49	22.20	4.02	20.00	23.00	25.00	
	Very dissatisfied	27	27.52	2.17	26.00	27.00	30.00	
**PSI**	Very satisfied	36	14.06	3.36	12.00	13.00	15.00	0.000 **
	Satisfied	62	18.05	5.09	14.00	18.00	21.25	
	Dissatisfied	49	25.73	6.85	19.50	26.00	31.00	
	Very dissatisfied	27	39.48	6.29	34.00	39.00	45.00	
**AC**	Very satisfied	36	11.11	2.12	10.00	11.00	13.00	0.000 **
	Satisfied	62	13.45	3.87	10.00	13.00	16.00	
	Dissatisfied	49	18.63	5.86	14.50	18.00	21.00	
	Very dissatisfied	27	30.07	6.48	28.00	31.00	34.00	
15–17 years								
**DSC**	Very satisfied	55	11.87	4.01	9.00	11.00	14.00	0.000 **
	Satisfied	53	18.11	4.81	14.00	18.00	21.00	
	Dissatisfied	70	22.71	3.12	21.00	23.00	25.00	
	Very dissatisfied	33	25.88	4.04	25.00	27.00	28.00	
**PSI**	Very satisfied	55	13.22	3.07	11.00	12.00	15.00	0.000 **
	Satisfied	53	17.94	4.98	14.00	17.00	21.00	
	Dissatisfied	70	25.34	6.63	21.00	25.50	30.00	
	Very dissatisfied	33	35.70	6.90	31.50	35.00	41.00	
**AC**	Very satisfied	55	10.16	2.08	8.00	10.00	11.00	0.000 **
	Satisfied	53	13.47	2.80	12.00	13.00	15.00	
	Dissatisfied	70	17.70	5.57	13.00	17.00	21.25	
	Very dissatisfied	33	27.61	5.68	25.00	28.00	31.50	
12–17 years								
**DSC**	Very satisfied	91	12.21	3.98	9.00	12.00	14.00	0.000 **
	Satisfied	115	17.51	4.73	14.00	18.00	21.00	
	Dissatisfied	119	22.50	3.51	20.00	23.00	25.00	
	Very dissatisfied	60	26.62	3.41	26.00	27.00	29.00	
**PSI**	Very satisfied	91	13.55	3.20	11.00	13.00	15.00	0.000 **
	Satisfied	115	18.00	5.02	14.00	18.00	21.00	
	Dissatisfied	119	25.50	6.70	21.00	26.00	30.00	
	Very dissatisfied	60	37.40	6.85	32.00	36.50	43.00	
**AC**	Very satisfied	91	10.54	2.14	9.00	10.00	12.00	0.000 **
	Satisfied	115	13.46	3.40	11.00	13.00	15.00	
	Dissatisfied	119	18.08	5.69	14.00	17.00	21.00	
	Very dissatisfied	60	28.72	6.13	25.00	29.00	33.00	

Scores of the DSC subscale items were reversed. ** *p* value < 0.001.

**Table 8 children-08-00448-t008:** Discriminant validity: association between the PIDAQ(A) and MI-S and MI-D.

Self-Rated Malocclusion Index (MI-S)												
Age Group	12–14 Years				15–17 Years				12–17 Years			
Quartile	Lower (Slight)	Upper (Severe)	Effect Size	*p* Value	Lower (Slight)	Upper (Severe)	Effect Size	*p* Value	Lower (Slight)	Upper (Severe)	Effect Size	*p* Value
**N**	62	62			60	67			122	129		
**PIDAQ** **Subscale**	**Mean (SD)**	**Mean (SD)**			**Mean (SD)**	**Mean (SD)**			**Mean (SD)**	**Mean (SD)**		
**DSC**	14.71 (4.98)	25.13 (3.48)	−2.43	**	13.15 (4.70)	25.04 (3.48)	−2.88	**	13.94 (4.88	25.08 (3.46)	−2.63	**
**PSI**	15.73 (4.70)	32.24 (9.36)	−2.23	**	14.23 (4.03)	31.01 (8.25)	−2.58	**	14.99 (4.43)	31.60 (8.79)	−2.39	**
**AC**	11.98 (3.05)	24.53 (7.84)	−2.11	**	10.70 (2.34)	23.60 (7.27)	−2.39	**	11.35 (2.79)	24.05 (7.54)	−2.23	**
**Investigator-Rated Index (MI-D)**												
**Age Group**	**12–14 Years**				**15–17 Years**				**12–17 Years**			
**Quartile**	**Lower (Slight)**	**Upper (Severe)**	**Effect Size**	***p* Value**	**Lower (Slight)**	**Upper (Severe)**	**Effect Size**	***p* Value**	**Lower (Slight)**	**Upper (Severe)**	**Effect Size**	***p* Value**
**N**	56	46			65	84			121	130		
**PIDAQ** **Subscale**	**Mean (SD)**	**Mean (SD)**			**Mean (SD)**	**Mean (SD)**			**Mean (SD)**	**Mean (SD)**		
**DSC**	15.30 (4.74)	21.24 (6.29)	−1.07	**	15.65 (5.98)	21.37 (6.15)	−0.94	**	15.49 (5.42)	21.32 (6.18)	−1.00	**
**PSI**	17.20 (5.54)	26.48 (9.83)	−1.16	**	17.51 (7.17)	24.50 (9.55)	−0.83	**	17.36 (6.44)	25.20 (9.66)	−0.96	**
**AC**	13.05 (3.15)	19.15 (7.88)	−1.02	**	13.74 (6.09)	17.81 (7.64)	−0.59	**	13.42 (4.95)	18.28 (7.73)	−0.75	**

Scores of the DSC subscale items were reversed. ** *p* value < 0.001.

**Table 9 children-08-00448-t009:** Discriminant validity: association between the PIDAQ(A) and need for orthodontic treatment.

PIDAQ Variables	Need Braces	N	PIDAQ Scores					*p* Value
			Mean	SD	Quartiles			
					Lower	Middle	Upper	
12–14 years								
**DSC**	Yes	81	23.96	4.20	22	24	27	0.000 **
	No	93	15.08	4.71	11.50	15	19	
**PSI**	Yes	81	29.61	9.70	22	29	38	0.000 **
	No	93	16.69	5.28	13	15	19	
**AC**	Yes	81	22.28	8.24	15	20	30	0.000 **
	No	93	12.41	3.17	10	12	15	
15–17 years								
**DSC**	Yes	118	22.89	4.32	20	23	26	0.000 **
	No	93	14.58	5.56	10	14	20	
**PSI**	Yes	118	26.64	8.74	19.75	27	33	0.000 **
	No	93	15.98	6.25	12	14	18	
**AC**	Yes	118	19.68	7.15	14	18	25	0.000 **
	No	93	11.84	4.03	9	11	13	
12–17 years								
**DSC**	Yes	199	23.33	4.29	21	24	27	0.000 **
	No	186	14.83	5.15	11	14	19	
**PSI**	Yes	199	27.85	9.24	21	28	34	0.000 **
	No	186	16.33	5.78	12	15	19	
**AC**	Yes	199	20.74	7.70	14	19	27	0.000 **
	No	186	12.12	3.63	10	11.50	14	

Scores of the DSC subscale items were reversed. ** *p* value < 0.001 for all subscales with all age groups.

**Table 10 children-08-00448-t010:** Discriminant validity of the PIDAQ(A) with IOTN-DHC.

PIDAQ Variables	Occlusal Traits	N	PIDAQ Scores					*p* Value
			Mean	SD	Quartiles			
					Lower	Middle	Upper	
12–14 Years								
**DSC**	No/Little	22	13.18	4.12	10.75	13.50	16	0.000 **
	Moderate	86	16.76	5.12	12.75	17	21.25	
	Great	61	24.30	4.30	22	25	28	
	Very great	5	26.00	3.67	23	27	28.50	
**PSI**	No/Little	22	14.77	4.08	12.75	13.50	16.50	0.000 **
	Moderate	86	18.19	5.90	13	17	22	
	Great	61	30.33	9.27	22.50	31	38	
	Very great	5	42.60	7.16	36.50	43	48.50	
**AC**	No/Little	22	11.59	2.68	10	10.50	14	0.000 **
	Moderate	86	13.30	4.06	10	13	15	
	Great	61	22.89	7.88	16	21	30	
	Very great	5	32.80	5.50	28	31	38.50	
15–17 years								
**DSC**	No/Little	43	12.84	4.96	9	12	15	0.000 **
	Moderate	100	18.28	5.16	13.25	19	22	
	Great	59	24.22	4.10	22	25	27	
	Very great	9	27.56	2.01	26	28	29.50	
**PSI**	No/Little	43	14.30	4.16	11	13	16	0.000 **
	Moderate	100	19.08	6.21	14	18	23.75	
	Great	59	30.25	8.64	25	32	36	
	Very great	9	35.78	7.41	31	34	43.50	
**AC**	No/Little	43	10.79	2.82	9	10	12	0.000 **
	Moderate	100	13.82	3.99	11	13	16	
	Great	59	22.83	7.41	17	24	29	
	Very great	9	25.56	6.77	21	25	30.50	
12–17 years								
**DSC**	No/Little	65	12.95	4.67	10	12	16	0.000 **
	Moderate	186	17.58	5.18	13	18	22	
	Great	120	24.26	4.19	22	25	27	
	Very great	14	27.00	2.69	26	27	29.25	
**PSI**	No/Little	65	14.46	4.11	11	13	16	0.000 **
	Moderate	186	18.67	6.07	13.75	18	23	
	Great	120	30.29	8.93	24	31	36	
	Very great	14	38.21	7.82	31	39	44.50	
**AC**	No/Little	65	11.06	2.78	9	10	13	0.000 **
	Moderate	186	13.58	4.02	10	13	16	
	Great	120	22.86	7.62	16.25	21	29.75	
	Very great	14	28.14	7.10	24.75	27.50	33.75	

Scores of the DSC subscale items were reversed. ** *p* value < 0.001.

**Table 11 children-08-00448-t011:** Floor and/or ceiling effects of the PIDAQ(A).

PIDAQ	12–14 Years (N)		15–17 Years (N)		12–17 Years (N)	
Subscale	% Floor	% Ceiling	% Floor	% Ceiling	% Floor	% Ceiling
DSC	0.6	4.0	1.9	1.4	1.3	2.6
PSI	3.4	1.1	4.7	0.5	4.2	0.3
AC	4.6	1.1	7.6	0.5	6.2	0.5

## Data Availability

Not applicable.

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
