# Peer review of "Validity and Reliability of the Arabic Version of the Psychosocial Impact of Dental Aesthetics Questionnaire for Yemeni Adolescents"

_children, 2021, doi:10.3390/children8060448_

Round 1

Reviewer 1 Report

The manuscript presents a thorough and comprehensive analysis of the Psychosocial Impact of Dental Aesthetics Questionnaire (PIDAQ) into Arabic version [PIDAQ(A)]. The  study goes well beyond a linguistic analysis and carries out a very detailed qualitative inter-view, comprehensibility assessment, and psychometric validations revision -for validity and reliability-. The results are exhaustively presented in form of tables and explained point by point in the manuscript. The use of sub-sections and a final corollary of the results woud increase reader's understanding of the text. The first sentence of the discussion would be better fitted to conclude this corollary.

Reviewer 2 Report

Dear Authors,

The manuscript entitled "Validity and Reliability of the Arabic Version of the Psychosocial Impact of Dental Aesthetics Questionnaire for Yemeni Adolescents" is an interesting study. However, I want to suggest some improvements and changes.

Introduction

I suggest to describe the IOTN index (mentioned in line 175) in the introduction part.

Line 72-73 please rewrite, sounds too colloquial.

"those" line 76 and 79 please change, maybe patients or persons.

Materials and Methods

Lines 97-124 please write it in a shorter manner it is to descriptive and to long, I suppose your purpose was to explain it step by step, but in my opinion it could be done in a shorter and an easier to follow manner. For example: "During the meeting, minor changes were made to the back translations and relevant changes were also made to the draft..." it sounds more like a novel not a scientific article.

Line 167 "in the classroom setting" again as above mentioned

Statistical analysis

It is so long and it is difficult to follow this part. Please remove the statistical descriptions like for example: 191-193, 206-209-it sound like teaching the reader the statistical methods. You wrote one and a half page of statistical analysis, it has the length of your discussion part, I do not think it is the correct approach. I hope you will find the way to make the statistical part more catchy without loosing its major points.

In the Materials and Methods part please add the ethical approval (I know you mentioned it at the end of the article, but as the important issue it should be stated also in the text of the same article).

Results

Lines 269-271, in the Methods part, you did not mention about the division into two age groups .

Lines 355-Table 8- please make it more legible, improve editing.

Discussion

Line 379 Terwee at al. instead of al,

Line 455 has not shown instead of hasn't shown

In order to make the discussion part more complex or to expand the introduction part to the related fields, like the attractiveness of a person despite some dental/facial defects. You may find the below mentioned article as the worth mentioning. According to some studies people express their solidarity and sympathize with persons affected even by clefts. This approach may become interesting for the readers of your article.

Sycinska-Dziarnowska, M.; Stepien, P.; Janiszewska-Olszowska, J.; Grocholewicz, K.; Jedlinski, M.; Grassi, R.; Mazur, M. Analysis of Instagram® Posts Referring to Cleft Lip. Int. J. Environ. Res. Public Health 202017, 7404. https://doi.org/10.3390/ijerph17207404

I admire your huge effort. Congratulations.

Kind regards,

Your Reviewer

Round 2

Reviewer 2 Report

Dear Authors,

In my opinion, after the corrections you have made, the article is suitable for the publications in the Journal. Thank you for the clear and made step by step explanations.

Congratulations!

Kind regards